# BALAUR: Language Model Pretraining with Lexical Semantic Relations

**Andrei Mircea**
Mila/McGill University
`mirceara`
`@mila.quebec`

**Jackie C. K. Cheung**
Mila/McGill University
`jcheung`
`@cs.mcgill.ca`

## Abstract

Lexical semantic relations (LSRs) characterize meaning relationships between words and play an important role in systematic generalization on lexical inference tasks. Notably, several tasks that require knowledge of hypernymy still pose a challenge for pretrained language models (LMs) such as BERT, underscoring the need to better align their linguistic behavior with our knowledge of LSRs. In this paper, we propose BALAUR, a model that addresses this challenge by modeling LSRs directly in the LM's hidden states throughout pretraining. Motivating our approach is the hypothesis that the internal representations of LMs can provide an interface to their observable linguistic behavior, and that by controlling one we can influence the other. We validate our hypothesis and demonstrate that BALAUR generally improves the performance of large transformer-based LMs on a comprehensive set of hypernymy-informed tasks, as well as on the original LM objective. Code and data are made available at `github.com/mirandrom/balaur`.

## 1 Introduction

Pretrained language models (LMs) trained on ever-increasing compute and data have achieved state-of-the-art performance on a wide variety of NLP benchmarks. However, they are still known to struggle on certain tasks, notably those involving reasoning and world knowledge (Liu et al., 2021). In particular, previous work has found that these models make errors on tasks involving lexical semantic relations (LSRs) such as hypernymy: notably cloze completions such as "*A fox is a type of* ___", where the hypernym "*canine*" is a valid completion (Ettinger, 2020); and monotonicity-based inferences such as "*Bob saw a fox*" entailing "*Bob saw a canine*" (Geiger et al., 2020).

This represents an important limitation of current LMs. Besides the specific classes of inferences in which they are involved, LSRs are crucial because

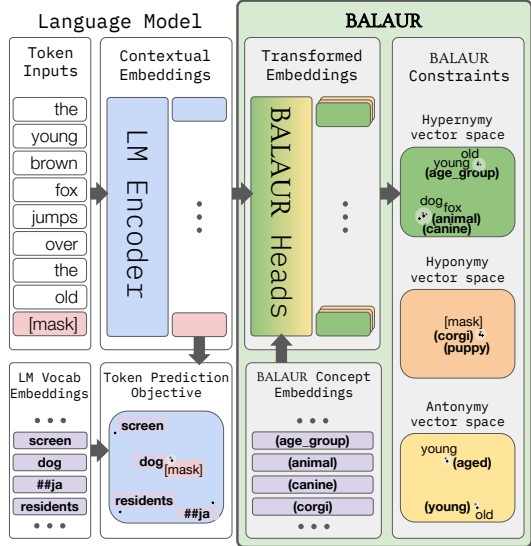

Figure 1: BALAUR learns to transform LM hidden states (contextual token embeddings) in LSR-specific vector spaces, modeling relatedness as similarity constraints.

they provide a dimension along which LMs can perform class-based generalization to new events in a sample-efficient manner. For example, knowing that a Xoloitzcuintli is a dog breed allows a model to infer many plausible properties about them, even without many occurrences of this word in training. This improved modelling of LSRs could in turn improve generalization and data efficiency in LMs.

We propose BALAUR (Figure 1), an approach to Transformer LM pretraining which directly models LSRs in the latent representations of the LM. BALAUR consists of a modular neural architecture with multiple heads, each head modeling a distinct LSR (e.g. hypernymy, synonymy, or antonymy). Each LSR is modelled as a learned vector transformation, with constraints injected in the resulting vector space to reflect structural properties of the LSR. Concretely, a BALAUR head transforms contextual token embeddings (e.g. `fox`) and static concept embeddings (e.g. `(canine)`) such that related pairs are similar in the corresponding LSR vector space (e.g. hypernymy).

Previous work has proposed to integrate LSRs with LMs, for example augmenting LM pretraining with hypernym class prediction (Bai et al., 2022). However, these approaches do not encode known structural properties of LSRs, modeling these indirectly without accounting for e.g. hypernymy being transitive, or antisymmetric with hyponymy.

In contrast, our method draws inspiration from work on semantic specialization, which models LSRs as constraints in the vector space of static word embeddings (e.g. Mrkšić et al., 2017). We show how similar insights can be applied to modern Transformer LM architectures such that the resulting LM's hidden states have inductive biases useful for LSR-informed tasks. More broadly, we demonstrate that our method creates an interface between the model's latent representations and its linguistic behaviour, and that we can control the latter by injecting LSR information into the former.

We evaluate BALAUR on several tasks which require knowledge of hypernymy. These include cloze completion, monotonicity-informed natural language inference (NLI) , and finetuning efficiency on the previous two tasks in a transfer learning setting. We find that BALAUR generally improves performance on these hypernymy-informed tasks, as well as the on the original LM objective.

**Contributions**

• We introduce BALAUR, a method aimed at improving the generalization of pretrained language models on tasks involving lexical semantic relations, specifically hypernymy and hyponymy.

• By modeling hypernymy and other lexical semantic relations in the hidden states of language models during pretraining, BALAUR consistently improves performance on language modeling and in a comprehensive set of hypernymy-informed tasks.

• Our evaluation brings together previous work on evaluating hypernymy in language models, providing a comprehensive view of how well hypernymy is captured in the linguistic behavior of LMs on tasks involving prompt completion, natural language inference and transfer learning.

• Finally, as part of this evaluation, we create and HYPCC, a dataset of hypernymy-informed cloze completion prompts improving on previous datasets with a better coverage of hypernymy and hyponymy. We also identify important challenges in creating such datasets from lexical resources.

## 2 Related Work

### 2.1 LM Pretraining with Hypernymy

Incorporating LSRs into LM pretraining, particularly hypernymy, has been approached from different angles. Lauscher et al. (2020) create supplemental training instances consisting of two words, where the model must predict whether they are semantically related using the next sentence prediction objective of Devlin et al. (2019). In contrast to our work, this approach combines synonymy, hypernymy and hyponymy into one relation and requires a large number of additional training examples during pretraining. Levine et al. (2020) avoid the need for additional training data by modifying the LM objective to jointly predict a word's supersense in addition to the word itself, while Bai et al. (2022) create a curriculum where LMs learns to predict a word's hypernym before predicting the word itself. However, these methods aim to improve LMs more broadly and do not target specific hypernymy-informed tasks or attempt to disentangle hypernymy from other relations during pretraining. In contrast, our work presents a novel method based on semantic specialization, where hypernymy and other LSRs are jointly modeled in the latent representations of LMs to improve performance on targeted evaluations of hypernymy. To the best of our knowledge there has been no work successfully using finetuning to incorporate task-agnostic knowledge of LSRs into LMs such as BERT, and adapting our method to finetuning consistently led to catastrophic forgetting.

### 2.2 Lexical Semantic Specialization

Prior to the advent of pretrained LMs, distributional word embeddings were augmented with LSRs using a class of techniques known as semantic specialization (Yu and Dredze, 2014; Glavaš and Vulić, 2018; Vulić et al., 2018, i.a.). These methods learn a transformation of the original distributional vector space that better captures relational knowledge such as LSRs, modeling relatedness as constraints in the resulting vector space. Our work most closely resembles that of Arora et al. (2020) which learns multiple relation-specific subspaces in the original vector space of word embeddings, and Gajbhiye et al. (2022) which use BERT-based bi-encoders to predict whether commonsense concept-property pairs are related based on the similarity of their transformed embeddings.

## 2.3 Evaluating Lexical Semantics in Models

There have been various approaches at evaluating how well a model's representations capture lexical semantics. In particular, there exists a plethora of intrinsic evaluations which probe representations directly, particularly for hypernymy relations. These typically take the form of relation prediction between term-pairs (e.g. Bordea et al., 2015; Santus et al., 2015; Bordea et al., 2016; Shwartz et al., 2016) with variants such as hypernym wordsense prediction (Espinosa-Anke et al., 2016), graded relation prediction (Vulić et al., 2017), hypernym retrieval from corpora (Camacho-Collados et al., 2018), and probing of LM representations (Vulić et al., 2020). However, such probing methods suffer from several limitations in the context of LMs, discussed by Rogers et al. (2020). Notably, probes tell us only what information can be recovered from LM representations, not how (or even if) the LM uses it in practice (Tenney et al., 2019). Because of these limitations and the fact that our approach explicitly models LSRs in LM representations, we do not conduct such intrinsic evaluations. In contrast, extrinsic evaluations measure LM performance on downstream inference tasks requiring knowledge of hypernymy (e.g. Geiger et al., 2020; Rozen et al., 2021). A fundamental challenge for such approaches is understanding whether performance is attributable to a model's learned representations or to finetuning. To address this, recent work has evaluated pretrained LMs in zero-shot prompt completion and finetuning efficiency (Talmor et al., 2020). Building on this work Ettinger (2020), Ravichander et al. (2020) and Hanna and Mareček (2021) demonstrate that modern LMs fail to generalize systematically on zero-shot cloze-style prompts informed by hypernymy. Our paper builds on this line of work in two ways. First, we present HYPCC, a dataset of cloze prompts with better coverage of hypernymy and hyponymy. Second, we bring together inference and prompt completion tasks into a comprehensive evaluation of hypernymy in LMs.

## 2.4 Lexical and Distributional Semantics

More broadly, these lines of work explore the interplay between lexical and distributional semantics, specifically how the first (in the form of LSRs) helps inform the second (in the form of training and evaluating LMs or word embeddings). Conversely, there is a rich body of work that has attempted to inform lexical semantics with distributional semantics. Of particular relevance to our work is the extraction from corpus data of hypernymy (Caraballo, 1999; Snow et al., 2004) and meronymy (Poesio et al., 2002) relations, typically based on Hearst patterns (Hearst, 1992). Similarly, Mohammad et al. (2008) leverage the co-occurrence hypothesis (Charles and Miller, 1989) to identify antonymy. Bridging the gap between lexical and distributional semantics, there is work like Agirre et al. (2009) which combines both approaches, noting that while distributional methods help alleviate out-of-vocabulary issues in lexical resources, they struggle to distinguish semantic similarity from relatedness. Our work attempts to address this issue, explicitly modeling LSRs in LM representations so they can be distinguished.

## 3 The BALAUR Head Architecture

In this section, we present the neural architecture of BALAUR heads, describing how they model LSRs in the hidden states of LMs, how this is translated into an optimizable loss function, and how they interface with LMs during pretraining.

**Assumptions** We take the term (neural) language model (LM) to refer to a neural network that predicts a token given its context, including commonly used masked LMs like BERT (Devlin et al., 2019) and auto-regressive LMs like GPT (Radford et al., 2018). These LMs encode a sequence of tokens into latent representations known as hidden states or contextualized token embeddings $T$, using these as inputs to a classification head that predicts the target token. Meanwhile, we represent a lexical semantic relation $R$ as a set of related lexical item pairs $(X_i \rightarrow X_j) \in R$, noting that LSRs can be directed, e.g. $(corgi \rightarrow dog)$ resides in hypernymy while $(dog \rightarrow corgi)$ resides in hyponymy.

## 3.1 Modeling Lexical Semantic Relations

Our goal is to model LSRs in $T$, where $T_i$ is the LM hidden state for token $i$. However, modeling LSRs between pairs of in-context tokens is challenging because related tokens often do not co-occur in the same context. One solution would be to model LSRs for related token pairs across different contexts, however this becomes intractable as the number of possible context pairs grows combinatorially.

Instead, we model LSRs as $(T_i \rightarrow C_j) \in R$, where $C$ is a set of context-independent concept embeddings learned during pretraining. Modeling

LSRs between token-concept pairs not only addresses the issue of related token co-occurrence, but also enables us to capture concepts that would otherwise fall outside the model's vocabulary.

For a given relation $R$, the corresponding BALAUR head learns to transform $T$ and $C$ such that related token-concept pairs are similar in the resulting relation-specific vector space. We implement these as two-layer neural networks with GELU activation functions (Hendrycks and Gimpel, 2020):

$$
\begin{aligned}
\underset{t \times b}{\mathrm{T}^R} &= \underset{b \times b}{W^R} \left( \mathrm{GELU}( \underset{t \times d}{T} \times \underset{d \times b}{W^{R,T}} + \underset{1 \times b}{B^{R,T}} )\right), \\
\underset{c \times b}{\mathrm{C}^R} &= \underset{b \times b}{W^R} \left( \mathrm{GELU}( \underset{c \times d}{C} \times \underset{d \times b}{W^{R,C}} + \underset{1 \times b}{B^{R,C}} )\right),
\end{aligned} \quad (1)
$$

where $t$ and $c$ are the number of token and concept embeddings, and $d$ and $b$ are their original and transformed dimensionalities. $W^R$, $W^{R,T}$, $W^{R,C}$, $B^{R,T}$, $B^{R,C}$ are learned projection and bias matrices that parameterize the transformations for $R$.

These learned transformations enable BALAUR heads to model and disentangle multiple LSRs in the vector space of $T$, i.e. ensuring a token's related concepts can be predicted from its contextualized embedding, distinguishing across different relations. Moreover, by parametrizing LSRs as learned transformations, our approach can model LSRs inductively; i.e. generalize from instances of related pairs to a functional representation that can extrapolate to unseen pairs (Vulić et al., 2018).

### 3.2 Optimizing a BALAUR Head

To translate our similarity constraint into a learning objective, we adapt the supervised contrastive loss of Khosla et al. (2020) which maximizes the inner product similarities $S$ between each related token-concept pair $(i, j)$, while minimizing it for unrelated pairs $(i, k)$. Optimizing this loss thus enables us to predict a token's related concepts from its contextual embeddings, encoding the corresponding LSR in the LM's hidden states:

$$
\mathcal{L}^R = \frac{1}{|R|} \sum_{(i,j)}^{R} - \log \frac{\exp\left(S_{i,j}^R\right)}{\sum_{k<c} \exp\left(S_{i,k}^R\right)} \quad (2)
$$

$$
\underset{t \times c}{\mathrm{S}^R} = \underset{t \times b}{T^R} \times \underset{c \times b}{(C^R)^T} \quad (3)
$$

where $i$ indexes the set of token embeddings, while $j$ and $k$ index the set of concept embeddings.

### 3.3 Interfacing with Language Models

During LM pretraining, $T$ is computed in the forward pass and used as input to the LM's classification head for token prediction. Each BALAUR head also takes $T$ as input, along with concept embeddings $C$ and relation-specific sets of indices $(i, j)$ — where $i$ indexes $T$ and the corresponding token in the training batch, while $j$ indexes a concept in $C$ related to $T_i$ by the corresponding relation $R$. Each head then computes its loss $\mathcal{L}^R$ and these are averaged before being added to the LM loss.

## 4 Method

In this section, we detail our methods for LM pretraining with BALAUR, using LSRs and concepts extracted from WordNet. We also present the architecture and hyperparameters for the LM in our experiments, a variant of BERT$_{\mathrm{LARGE}}$ suitable for academic budgets. While our experiments are limited to masked language modeling and LSRs, our method can be extended to autoregressive language modeling and other forms of relational knowledge.

### 4.1 Extracting LSRs from WordNet

As a first step, we extract related token-concept pairs for hypernymy, hyponymy, antonymy and synonymy from WordNet's noun hierarchy (Miller, 1995). To do this, we begin by mapping the model's vocabulary to corresponding WordNet synsets (referred to throughout this paper as concepts) using NLTK (Bird and Loper, 2004). For example, the token `dog` maps to the concept of a pet dog **(dog.n.01)**, or a hot dog **(frank.n.02)**.

Next, using the resulting set of concepts, we extract related concept-concept pairs from WordNet and convert these to token-concept pairs. For example **(canine.n.01)** is a hypernym of **(dog.n.01)**, while **(sausage.n.01)** is a hypernym of **(frank.n.02)**; but both are extracted as hypernyms of the token `dog`. To improve coverage of WordNet, we consider multi-hop hypernymy up to depth 3, such that e.g. **(animal.n.01)** is extracted as a hypernym of both `dog` and `canine`. The resulting set of token-concept pairs contains $15,612$ unique concepts.

Manual sampling and inspection of the resulting pairs revealed several known issues associated with WordNet, including inaccurate lemmatization (McCrae et al., 2019), and too fine-grained word senses (McCarthy, 2006), further discussed in §7. To help address these potential sources of noise in BAL-

AUR, we filter tokens, concepts and token-concept pairs using the criteria described in A.1.1.

## 4.2 Incorporating BALAUR into Pretraining

We then use these token-concept pairs to optimize a BALAUR head for each LSR throughout LM pretraining. First, we randomly initialize $C$ as a $15612 \times 768$ embedding layer. Second, training examples are annotated with relation-specific sets of indices $(i, j)$, where $i$ indexes a token in the training sequence and $j$ indexes a related concept in $C$. Lastly, the hidden states $T$ are computed in the LM's forward pass, and fed into each BALAUR head, along with $C$ and the sets of indices $(i, j)$, to compute $\mathcal{L}^R$ as described in §3.3, using a transformed dimensionality $b$ of 768. To prevent sequentially iterating over BALAUR heads, we adopt the parallelization technique from multi-head attention (Vaswani et al., 2017). Specifically, we learn one set of transformations (1) but multiply $b$ by $|R|$ so the resulting transformed vector space can be partitioned across relations. To reduce memory overhead, we only input the subset of $T$ containing LSRs into BALAUR, reindexing $i$ on this subset.

## 4.3 Language Model Pretraining Setup

Our LM architecture, pretraining procedure, and hyperparameters are based on 24hBERT (Izsak et al., 2021) which enables rapid pretraining with limited resources, while reaching comparable performance with the original BERT models (Devlin et al., 2019). Specifically, we pretrain a BERT$_{\text{LARGE}}$ architecture to perform masked language modeling (MLM) on 128-token sequences for 25,000 steps with a batch size of 4,096 and using 16-bit precision. We optimize using AdamW (Loshchilov and Hutter, 2019) and a peak learning rate of 2e-3 with warm-up over the first 1,500 steps and linear decay. The pretraining data is a snapshot of English Wikipedia from 2022-03-01, and BookCorpusOpen (Bandy and Vincent, 2021), with $0.5\%$ withheld for validation. These datasets were downloaded from and preprocessed with the `datasets` library (Lhoest et al., 2021) which provided licenses such as CC-BY-SA 3.0 and GFDL for Wikipedia.

## 5 Evaluating BALAUR Language Models

In this section, we evaluate whether BALAUR heads can improve performance on tasks that are informed by LSRs, specifically hypernymy and hyponymy. To this end, we compare BERT$_{\text{LARGE}}$ mod-

els that were pretrained with and without BALAUR heads. Throughout this section, we refer to these as BERT+BALAUR and BERT (OURS) respectively.

Drawing from the observation that LSRs constrain language production, understanding, and learning in humans (Nagy and Gentner, 1990; Fass, 1993); we assemble a gauntlet of hypernymy-informed evaluations that broadly mirror these three capabilities, providing a comprehensive view of different ways LMs can capture hypernymy in their linguistic behavior:

**Language modeling** (§5.1): to complement our evaluation, we verify the effect BALAUR has on the original LM objective, with particular attention to performance on tokens with hypernymy relations.
**Prompt completion** (§5.2): models must predict the correct token given a cloze-style prompt describing a hypernymy or hyponymy relation, e.g. "*a dog is a type of* `[mask]`".
**Monotonicity NLI** (§5.3): models must predict whether a sentence entails another, when hypernymy and monotonicity determine entailment, e.g. "*drive a taxi*" entails "*drive a car*".
**Finetuning Efficiency** (§5.4): we compare how efficiently models transfer-learn when finetuned on the two previous tasks, disambiguating what is learned during pretraining versus during finetuning.

## 5.1 Language Modeling

In Table 1, we see that incorporating BALAUR into the LM pretraining procedure of Izsak et al. (2021) increases both negative log likelihood (NLL) and mean reciprocal rank (MRR) for the original masked language modeling objective. We observe similar improvements when masking random tokens as when masking only tokens with LSRs, indicating the improvements introduced by BALAUR extend beyond the modeling of LSRs. Lastly, we note that improvements in the original MLM objective begin early and are consistent throughout pretraining, as seen in Figure 2.

| Model | Random Tokens | | Lsr Tokens | |
|---|---|---|---|---|
| | NLL | MRR | NLL | MRR |
| BERT (OURS) | 1.659 | 0.733 | 3.359 | 0.482 |
| BERT+BALAUR | 1.587 | 0.743 | 3.201 | 0.503 |
| $\Delta(\%)$ | 4.3 | 1.4 | 4.5 | 4.1 |

Table 1: Validation MLM performance, shown for masking random tokens and for only masking tokens with LSRs (i.e. modeled by BALAUR during pretraining).

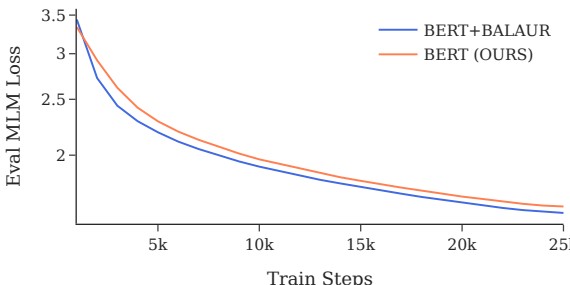

Figure 2: Validation MLM loss throughout pretraining.

## 5.2 Prompt Completion

**Task Description** We create HYPCC: a dataset of cloze-style prompts taking the form "In the context of hypernymy, a(n) $x$ is a type of $y$." where $x, y$ are hyponym-hypernym pairs of tokens in our model's vocabulary, and either is masked out to be predicted by the model. This evaluation builds on the work of Ettinger (2020) and Ravichander et al. (2020), which draws from human psycholinguistic tests to create cloze prompts. In contrast to previous work, our evaluation includes hypernyms beyond Fischler categories, evaluates hyponym prediction, considers tokens with multiple word senses, and includes clozes with multiple valid completions. The resulting dataset contains 17,556 hyponym-hypernym pairs; 5,217 hypernym prediction prompts; and 4,115 hyponym prediction prompts. We report additional details for the creation of HYPCC in A.1.2, and discuss several limitations in §7.2.

**Evaluation Method** In line with previous work, models are evaluated on HYPCC in a zero-shot manner (i.e. using masked language modeling to complete the cloze prompt); and performance measured with accuracy and mean reciprocal rank (MRR) for both the open and closed vocabulary settings. In the closed setting, metrics are calculated using only the set of possible hypernyms or hyponyms in HYPCC, while the open setting considers the model's entire vocabulary. Importantly, these metrics are adjusted to account for multiple valid completions in a prompt: ignoring other valid completions when computing a completion's rank (i.e. if a model's top three predictions are valid, the average accuracy will be 100% instead of 33%). To prevent a skewing of results by prompts with a larger number of completions, metrics are first averaged over completions, then averaged over prompts.

**Results and Discussion** In Table 2, we find that BALAUR improves performance on hypernymy-informed prompt completion across settings and metrics, even outperforming the original BERT_LARGE implementation of Devlin et al. (2019).

However, we note that both of our models struggle with ACC@1 when compared to †BERT_LARGE, despite general improvements of BALAUR over our baseline. A closer inspection of model predictions reveals that, similar to findings of Ettinger (2020), models often repeat the hypernym or hyponym in the context (e.g. predicting "*a daisy is a type of daisy*"). In Table 3, we find that our baseline pretraining procedure exacerbates this problem, explaining the discrepancy in ACC@1 performance.

Moreover, a qualitative analysis of selected clozes similar to Arora et al. (2020), shown in Table 4, suggests that BALAUR better disentangles hypernymy from other forms of semantic relatedness. These results agree with Agirre et al. (2009), who showed similar improvements combining lexical and distributional semantics in word embeddings.

It is also interesting to note that BALAUR spreads its probability mass more evenly across predictions, better capturing the one-to-many nature of hypernymy relations. However, we observe that many of the seemingly valid completions are not actually gold-standard completions in HYPCC. This is because HYPCC considers only direct hypernymy relations in WordNet, while several completions are indirect hypernymy relations or not in WordNet. We further discuss these limitations in §7.

| MODEL | CLOSED VOCAB | | OPEN VOCAB | |
|---|---|---|---|---|
| | ACC@1/5 | MRR | ACC@1/5 | MRR |
| *HYPERNYM PREDICTION* | | | | |
| †BERT_LARGE | 3.53 / 14.13 | 0.092 | **1.78** / 11.77 | 0.071 |
| BERT (OURS) | 5.18 / 18.61 | 0.121 | 0.88 / 14.72 | 0.080 |
| BERT+BALAUR | **5.31** / **19.65** | **0.128** | 1.60 / **15.44** | **0.089** |
| *HYPONYM PREDICTION* | | | | |
| †BERT_LARGE | **3.60** / 14.95 | 0.097 | **2.76** / 12.87 | 0.083 |
| BERT (OURS) | 2.69 / 12.22 | 0.081 | 2.03 / 10.65 | 0.069 |
| BERT+BALAUR | 3.49 / **17.91** | **0.110** | 1.85 / **14.56** | **0.084** |

Table 2: Zero-shot results on HYPCC. BALAUR generally improves performance across metrics when compared to a baseline BERT model with the same 24hBERT pretraining procedure, as well as the published checkpoint of †BERT_LARGE (Wolf et al., 2020). More extensive comparisons are included in §A.2.1.

| MODEL | HYPERNYM REPETITION | HYPONYM REPETITION |
|---|---|---|
| †BERT$_{LARGE}$ | 50.17 | 47.08 |
| BERT (OURS) | 87.81 | 64.20 |
| BERT+BALAUR | 69.59 | 69.38 |

Table 3: Rates of repetition on HYPCC. BALAUR reduces repetition for hypernym prediction, with comparable rates of repetition for hyponym prediction.

| In the context of hypernymy, a church is a type of `[mask]`. | | | | |
|---|---|---|---|---|
| BERT (OURS) | church 74.78 | **religion** 2.83 | structure 1.25 | building 1.11 | **worship** 0.86 |
| BERT+ BALAUR | church 27.33 | building 21.45 | structure 15.57 | place 2.41 | object 1.83 |
| In the context of hypernymy, a `[mask]` is a type of poem. | | | | |
| BERT (OURS) | poem 91.72 | **poet** 0.84 | poetry 0.55 | verse 0.50 | **word** 0.35 |
| BERT+ BALAUR | poem 66.23 | verse 3.80 | song 3.46 | poetry 2.47 | " " 1.67 |
| In the context of hypernymy, a volcano is a type of `[mask]`. | | | | |
| BERT (OURS) | volcano 88.30 | **lava** 1.59 | cone 1.11 | **rock** 0.94 | **eruption** 0.88 |
| BERT+ BALAUR | volcano 69.54 | mountain 13.27 | structure 2.55 | object 0.80 | **rock** 0.69 |

Table 4: Top-5 completions and probability percentages for selected clozes, showcasing how BALAUR can help disentangle hypernymy from other forms of semantic relatedness (related but invalid completions are bolded).

## 5.3 Monotonicity NLI

**Task Description** Our second evaluation is taken from Geiger et al. (2020), who create MoNLI: a challenge NLI dataset where entailment is determined by hypernymy. For instance, "*A man is talking to someone in a taxi*" entails "*A man is talking to someone in a car*". While models finetuned on SNLI (Bowman et al., 2015) perform well on such examples, they fail to generalize on examples where negation reverses entailment. For instance, "*A man is not talking to someone in a car*" now entails "*A man is not talking to someone in a taxi*". MoNLI is divided into PMoNLI and NMoNLI to distinguish between positive and negated examples.

**Evaluation Method** We adopt the evaluation procedure of Geiger et al. (2020), reporting test set accuracies for models finetuned on SNLI, and models also finetuned on MoNLI. We follow the NLI finetuning procedure of 24hBERT (Izsak et al., 2021) on which our model is based. However, we found that performance is sensitive to random seeds, so we report results averaged across 5 seeds.

**Results and Discussion** In Table 5, we replicate the results of Geiger et al. (2020), finding that models finetuned on SNLI only generalize to PMoNLI but fail completely on NMoNLI. Unexpectedly, we find BALAUR improves both SNLI and PMoNLI performance in this setting, suggesting some examples in SNLI also benefit from the representations learned with BALAUR pretraining.

| MODEL | SNLI | PMoNLI | NMoNLI |
|---|---|---|---|
| SNLI FINETUNING ONLY | | | |
| BERT (OURS) | 85.44 | 65.51 | 0.50 |
| BERT+BALAUR | 86.49 | 76.92 | 0.10 |
| SNLI + MoNLI FINETUNING | | | |
| BERT (OURS) | 85.43 | - | 48.90 |
| BERT+BALAUR | 86.38 | - | 56.50 |

Table 5: SNLI and MoNLI accuracies.

However, we conversely find that BALAUR degrades performance on the withheld test set of NMoNLI. While BALAUR may help LMs better capture hypernymy, the fact that it does not account for negation may help explain this result. Furthermore, visualizing performance across seeds in §A.2.3, we observe markedly larger variance on NMoNLI compared to SNLI and PMoNLI, making this result more difficult to interpret reliably.

## 5.4 Finetuning Efficiency

**Task Description** Our final evaluation reframes §5.2 and §5.3 not in terms of zero-shot or final performance, but in terms of performance throughout the finetuning of a pretrained model — as proposed by Talmor et al. (2020) in oLMpics. This approach was originally proposed because finetuning pretrained LMs makes it hard to disentangle what is captured in the pretrained representations from what is learned during finetuning. A key assumption underlying our use of this evaluation is that models which better capture hypernymy in their pretrained representations will be finetuned more efficiently (i.e. with better finetuned performance relative to finetuning steps, throughout finetuning).

**Evaluation Method** We finetune our models using the hyperparameters and finetuning procedures from Talmor et al. (2020) for prompt completion, and from Geiger et al. (2020) for MoNLI. This includes freezing model parameters for the prompt completion task (leaving only the language modeling head unfrozen); while unfreezing the entire model for the NLI task. We perform 5-fold cross-

validation with a 20% split, and average validation set results. Importantly, the splits are systematic to ensure that no hypernyms or hyponyms occur in both train and validation sets.

**Results and Discussion (Prompt Completion)** In Figure 3, we see that BALAUR's improvement on hypernym prediction extends throughout finetuning, indicating better transfer learning abilities. We show similar results for the hyponym subset of HYPCC in §A.2.2.

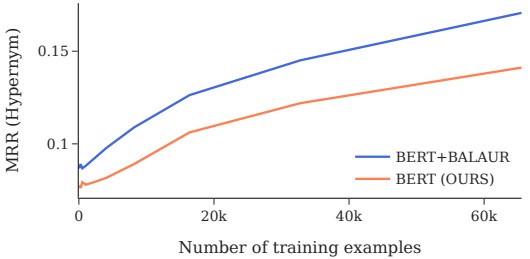

Figure 3: Average open-vocab MRR throughout finetuning on the hypernym prediction subset of HYPCC.

However, it is puzzling that performance remains relatively low despite extensive finetuning. A closer look at the outputs of the final model reveals that many of the erroneous entries in the model's top-10 open vocabulary predictions were in fact other classes in the HYPCC dataset (i.e. tokens from the closed vocabulary). In Figure 4, we quantify the class intrusion rate as the proportion of top-10 predictions which are both erroneous and a class in HYPCC, finding that it increases significantly throughout finetuning.

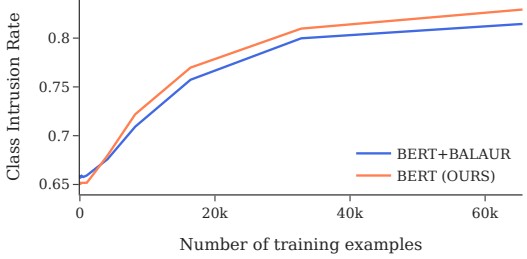

Figure 4: Average class intrusion rate throughout finetuning on the hypernym prediction subset of HYPCC.

One possible explanation is that models learn to predict indirect hypernyms or hyponyms not accounted for in HYPCC, similar to examples in Table 4. However, a manual inspection of model predictions showed that this was not often the case.

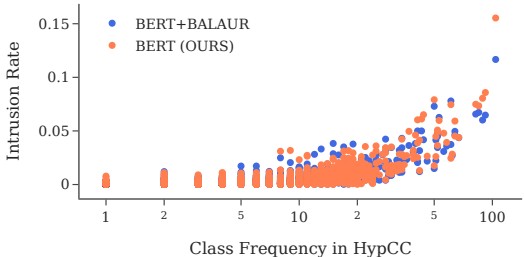

Figure 5: Average intrusion rate and frequency of classes in the final models finetuned on the hypernym prediction subset of HYPCC.

Instead, in Figure 5, we find that the intrusion rate of a class grows with its frequency in the finetuning dataset. Given that intrusion rates increase with finetuning and that frequent classes have higher intrusion rates, this suggests that LMs struggle to discriminate single token differences in prompts, and instead conflate learning signal across prompts with more frequent classes dominating.

**Results and Discussion (MoNLI)** In Figure 6, we observe similar results for MoNLI, indicating that BALAUR improves finetuning efficiency. In contrast to the results in §5.3, we also observe in Table 6 that BALAUR improves final performance on systematic validation splits for both PMoNL and NMoNLI. These improvements are consistent even when stratifying by BALAUR coverage of the hypernym and hyponym in a given MoNLI example, i.e. whether or not BALAUR models hypernymy or hyponymy relations for these tokens.

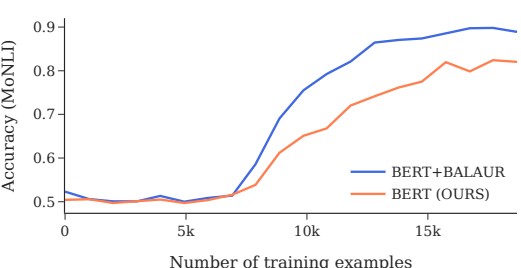

Figure 6: Average accuracy throughout finetuning.

|  | Overall Accuracy | Accuracy by BALAUR Coverage | | |
|---|---|---|---|---|
|  |  | Hyper+Hypo | Hyper Only | Neither |
| *PMoNLI* | | | | |
| BERT (OURS) | 82.14 | 80.69 | 85.07 | 58.33 |
| BERT+BALAUR | 86.78 | 86.19 | 88.27 | 72.22 |
| *NMoNLI* | | | | |
| BERT (OURS) | 80.31 | 78.00 | 81.81 | - |
| BERT+BALAUR | 93.01 | 91.44 | 94.02 | - |

Table 6: Final performance on MoNLI subsets, averaged over five systematic validation splits and stratified by coverage (subsets with no coverage are omitted).

# 6  Conclusion

In this work, we set out to align the linguistic behavior of LMs with our knowledge of LSRs and improve their performance on hypernymy-informed tasks. We presented BALAUR, an approach that aims to guide the linguistic behavior of LMs in such a way by modeling LSRs directly in their hidden states throughout pretraining.

Underlying this proposed approach was the hypothesis that LM latent representations can provide an interface to their linguistic behavior, and that controlling one can help guide the other. To verify our hypothesis, we characterized the effect of BALAUR on a series of evaluations, targeting several distinct ways in which LMs might capture hypernymy in their linguistic behavior.

Our findings show that BALAUR can robustly improve performance on diverse hypernymy-informed tasks, validating the effectiveness of our method while supporting our original hypothesis. Notably, we demonstrated that BALAUR also improves performance on the original language modeling objective, indicating our method's improvements are not limited to hypernymy-informed tasks and can extend to more general linguistic behavior. However, we found that aligning the linguistic behavior of LMs with BALAUR still poses several challenges. We further discuss these limitations in §7 and outline implications for future work.

More broadly, BALAUR is a general-purpose architecture for modeling relations in the latent representations of neural network models. While our work has focused on modeling LSRs in LM hidden states throughout pretraining, BALAUR can in principle be applied to different modalities, architectures, relations and optimization settings. How well our results and hypothesis generalize to such different settings remains an open question.

# 7  Limitations

## 7.1  Pretraining from scratch

Our work attempts to improve LMs by pretraining from scratch, adopting a relatively efficient pretraining approach geared towards academic budgets (Izsak et al., 2021). While our approach can in principle generalize to other LMs such as GPT (Radford et al., 2018), we could not pretrain these models due to limited computational resources.

Adapting our method to effectively finetune existing pretrained models could significantly reduce the compute and data required. While this approach seems more practical and accessible, in practice we found that effectively finetuning pretrained LMs to improve their linguistic behavior across a range of tasks without loss of generality is more difficult from an optimization perspective than pretraining a different model from scratch.

While previous work has succeeded in finetuning pretrained LMs without catastrophic forgetting (e.g. for downstream tasks (Chen et al., 2020), domain adaptation (Gururangan et al., 2020), and de-biasing (Gira et al., 2022)), these do not address the issue of incorporating external knowledge in pretrained LMs to guide their linguistic behavior across a variety of tasks. To the best of our knowledge, achieving this remains an open question.

## 7.2  Noise, bias and coverage in WordNet

When using knowledge bases such as WordNet, it is important to account for their inherent limitations. In particular, we identify three prevalent issues in WordNet that can negatively affect what LMs learn in our experiments.

First is the problem of noise. Due to issues with lemmatization, word sense granularity and idiomaticity, we found many questionable relations being extracted when creating training data for BALAUR and examples for HYPCC. For example, we find that the token `cat` is lemmatized to the concept **(cat-o'-nine-tails.n.01)**, implying `cat` has the hypernym **(whip.n.01)**. Conversely, word sense granularity can lead to questionable relations like `chair` **(professorship.n.01)** being a hyponym of `situation` **(position.n.06)**". Lastly, idioms like "taking a crack at something" can lead to (unlikely when taken out of context) relations like "crack" having the hypernym "endeavor". These limitations are exacerbated in our experiments, as we do not disambiguate word senses, considering all possible meanings of a given token instead.

Second is the issue of bias. We found WordNet to encode several harmful biases and stereotypes, either directly via harmful relations, or indirectly by including certain relations for some groups but not others. For example, "man" has hyponyms "bachelor", "officer" and "gallant"; in contrast to "mistress", "nurse" and "tease" for "woman". Despite removing these associations in our work, we want to note that these kind of biases can be difficult to

comprehensively account for when expressed as selective inclusion or omission of associations for different groups.

Lastly, is the related issue of coverage. Many concepts and relations are simply not expressed in WordNet; limiting the knowledge of LSRs that can be incorporated in LMs with this resource. This lack of coverage is exacerbated in our experiments, as we are limited to single token words (i.e. words in the model's vocabulary). Despite trying to alleviate this by also modeling extra-vocabulary concepts, effectively controlling the representations of multi-token expressions in LMs remains an open problem. We note that, due to its reliance on expert lexicographers, WordNet has had limited updates and developments to increase its coverage; this is in contrast to the open sourced English WordNet 2019 (McCrae et al., 2019). We suggest future work consider this resource to mitigate coverage issues.

## Acknowledgements

We would like to thank Samsung Electronics Co., Ldt. and Fonds de recherche du Québec for funding this research. We would also like to thank the Mila IDT team for computational resources and support. The authors acknowledge the material support of NVIDIA in the form of computational resources.

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

# A   Appendix

## A.1   Additional method details

### A.1.1   Filtering LSRs from WordNet

**Filtering Tokens** When mapping tokens to WordNet synsets using NLTK, we observed several potential sources of noise that could be addressed by filtering tokens. First, we observed that many tokens with 3 or fewer characters were often over-zealously lemmatized by NLTK as acronyms or abbreviations for unlikely synsets. For example `cat` may map to `computerized_tomography.n.01`, while `in` maps to `indium.n.01`, `indiana.n.01`, and `inch.n.01`. Originally, we attempted to filter any token with 3 or fewer characters, however our coverage of important concepts dropped significantly, so we limit ourselves to filtering tokens with 2 or fewer characters. We also filter out tokens which are wordpieces in the model vocabulary (e.g. tokens prefixed by "##" in the vocabulary of BERT, indicating these are not preceded by whitespace and occur in the middle of words) to ensure we only model LSRs for tokens that correspond to entire words. We also use a WordNet stoplist (Pedersen and Banerjee, 2009) to filter common function words that tend to be misrepresented by WordNet. Lastly, we limit ourselves to alphabetical tokens, as we found numerical and alphanumerical tokens to introduce a lot of noise.

**Filtering Synsets** Having filtered tokens, we then map each of these to all possible synsets using the NLTK interface to WordNet. However, we found the quality, coverage and ambiguity of annotations to vary significantly across synset types. To reduce noise, we filtered synset categories based on manual inspection. We first limit ourselves to noun synsets, and filter what we found to be particularly noisy categories: quantity, motive, shape, relation, and process. Furthermore, we found that despite filtering tokens from our stoplist, NLTK was still lemmatizing other tokens to synsets in the stoplist, so we further filter any synset whose identifiers are in the stoplist.

**Filtering Token-Concept Pairs** After mapping hypernymy, hyponymy, synonymy and antonymy relations between tokens and synsets, we filter synsets based on their coverage of our model's vocabu-

lary. Specifically, our goal is to avoid modeling LSRs for synsets that only relate to one item in our vocabulary, as these cannot provide any useful inductive bias to our model's representations of its vocabulary. We first keep any synsets which map to 2 or more tokens (i.e. capture synonymy). If a remaining synset has antonymous synsets, we keep it if both it and its antonym(s) have corresponding tokens in the model vocabulary. Lastly, if a remaining synset belongs in a hypernymy or hyponymy relation, we keep it even if it does not map to a token, as long as it relates to two or more hypernym or hyponym synsets that do. This enables us to indirectly model concepts not in the model vocabulary via co-hyponymy and co-hypernymy relations. Any remaining synset is removed, along with its related token-concept pairs. This filtering ensures that we model concepts relating to multiple tokens in our vocabulary and prevents the degenerate case where a concept is indistinguishable from a token.

### A.1.2   HYPCC dataset creation

To create HYPCC, we first extract related token-concept pairs using the same procedure outlined in §4.1 and A.1.1. One notable difference is that we only consider direct hypernyms, instead of multi-hop hypernyms up to depth 3. Furthermore, we filter tokens such that they occur in the two most frequent English LM vocabularies: `bert-base-uncased` and `gpt2`, as hosted by the transformers library (Wolf et al., 2020).

We then convert token-concept pairs to sets of token-token pairs, based on the concepts' surface forms which are present in our vocabularies. To convert these pairs to cloze-style prompts, we adopt the following template: "A(n) $X$ is a type of $Y$". We use the inflect library [1] to filter plural forms or determine the adequate article ("a" or "an"). While we do not account for uncountable nouns, we find that most prompts maintain their legibility.

Lastly, we found that several concepts and tokens were disproportionately represented in this dataset as a result of having multiple wordsenses or maintaining a high position in the WordNet hierarchy. These often lead to nonsensical prompts, which we attempted to filter out using a manually curated stoplist for tokens and concepts.

---

[1] https://github.com/jaraco/inflect

## A.2 Additional results

### A.2.1 Extended zero-shot results on HYPCC

| MODEL | CLOSED VOCAB | | OPEN VOCAB | |
|---|---|---|---|---|
| | ACC@1/5 | MRR | ACC@1/5 | MRR |
| HYPERNYM PREDICTION | | | | |
| BERT_BASE | 2.75 / 12.88 | 0.081 | 0.30 / 10.25 | 0.054 |
| BERT_LARGE | 3.53 / 14.13 | 0.092 | 1.78 / 11.77 | 0.071 |
| ROBERTA_BASE | 4.46 / 15.54 | 0.103 | 1.90 / 12.14 | 0.074 |
| ROBERTA_LARGE | 7.01 / 20.12 | 0.137 | 5.29 / 17.00 | 0.114 |
| BERT (ours) | 5.18 / 18.61 | 0.121 | 0.88 / 14.72 | 0.080 |
| BERT+BALAUR (ours) | 5.31 / 19.65 | 0.128 | 1.60 / 15.44 | 0.089 |
| HYPONYM PREDICTION | | | | |
| BERT_BASE | 1.99 / 11.89 | 0.073 | 1.39 / 10.42 | 0.061 |
| BERT_LARGE | 3.60 / 14.95 | 0.097 | 2.76 / 12.87 | 0.083 |
| ROBERTA_BASE | 2.94 / 12.06 | 0.080 | 2.24 / 9.92 | 0.066 |
| ROBERTA_LARGE | 3.89 / 12.90 | 0.091 | 3.37 / 11.55 | 0.081 |
| BERT (ours) | 2.69 / 12.22 | 0.081 | 2.03 / 10.65 | 0.069 |
| BERT+BALAUR (ours) | 3.49 / 17.91 | 0.110 | 1.85 / 14.56 | 0.084 |

Table 7: Zero-shot results on HYPCC across MLMs. We note that RoBERTa was trained on an order of magnitude more data than the models used in our experiments (16GB versus 161GB), which has a significant impact on downstream performance (Liu et al., 2019) and helps explain the discrepancy in performance. In contrast and in line with Nityasya et al. (2023), our main results aim to disentangle the effects of scaling data or compute.

### A.2.2 Extended finetuning results on HYPCC

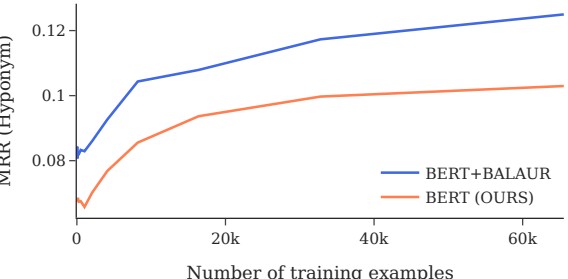

Figure 7: Average open-vocab MRR throughout finetuning on the hyponym prediction subset of HYPCC.

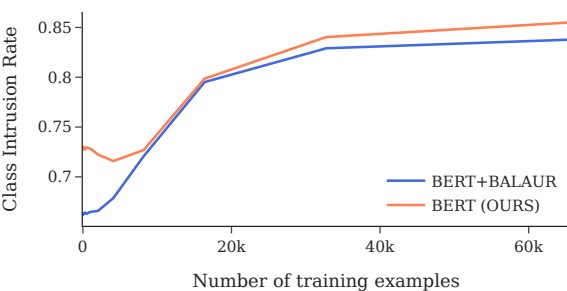

Figure 8: Average class intrusion rate throughout finetuning on the hyponym prediction subset of HYPCC.

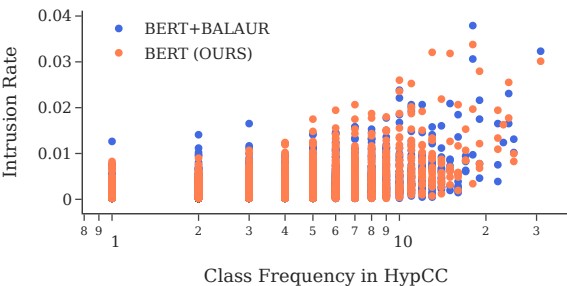

Figure 9: Average intrusion rate and frequency of classes in the final models finetuned on the hyponym prediction subset of HYPCC.

### A.2.3 MoNLI performance across random seeds

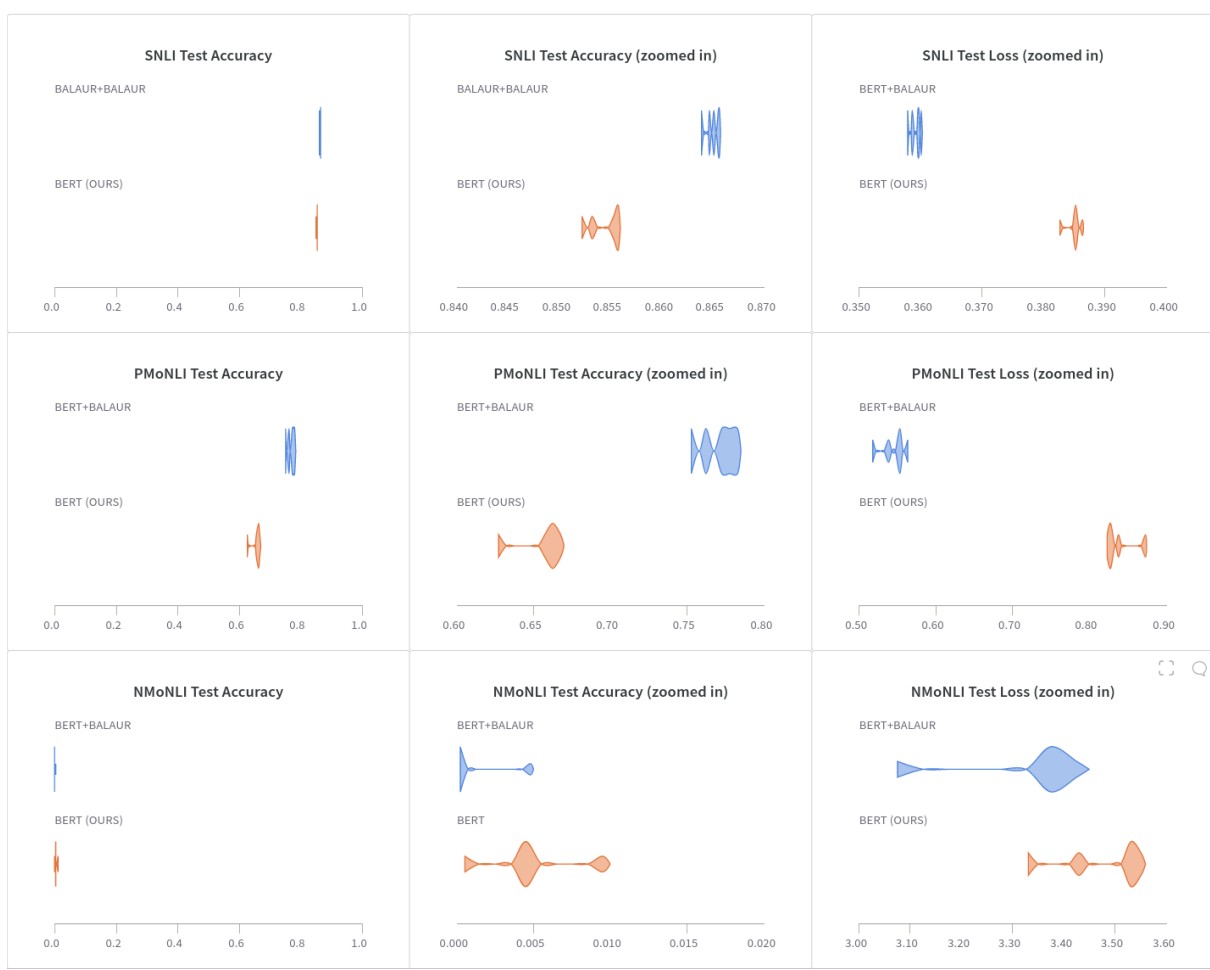

Figure 10: MoNLI performance across 5 seeds when finetuned only on SNLI.

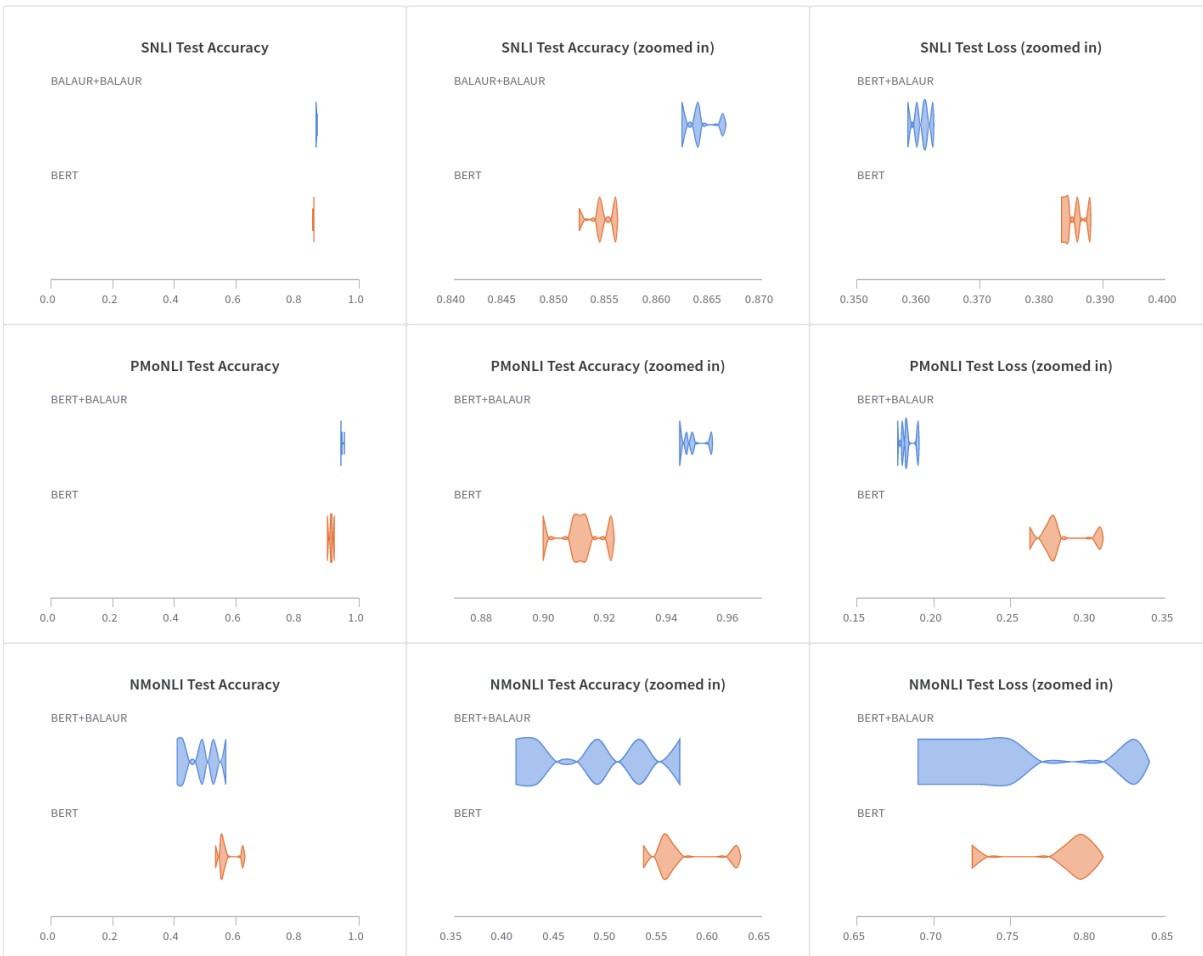

Figure 11: MoNLI performance across 5 seeds when
finetuned on SNLI and MoNLI.