# OpenReview forum: "Balaur: Language Model Pretraining with Lexical Semantic Relations"
_EMNLP/2023/Conference — EMNLP 2023 Findings_

### Official Review · Reviewer_8X5a · 2023-08-04

**Soundness:** 3

**Excitement:**

3: Ambivalent: It has merits (e.g., it reports state-of-the-art results, the idea is nice), but there are key weaknesses (e.g., it describes incremental work), and it can significantly benefit from another round of revision. However, I won't object to accepting it if my co-reviewers champion it.

**Missing References:**

While a bit dated, the below references should give an account of existing datasets and methods that have been used in the hypernymy modeling space in the past years.

Tasks and datasets that explicitly measure hypernymy

	TexEval 1 and 2
	Georgeta Bordea, Paul Buitelaar, Stefano Faralli, and Roberto Navigli. 2015. SemEval-2015 Task 17: Taxonomy Extraction Evaluation (TExEval). In Proceedings of the 9th International Workshop on Semantic Evaluation (SemEval 2015), pages 902–910, Denver, Colorado. Association for Computational Linguistics.

	Georgeta Bordea, Els Lefever, and Paul Buitelaar. 2016. SemEval-2016 Task 13: Taxonomy Extraction Evaluation (TExEval-2). In Proceedings of the 10th International Workshop on Semantic Evaluation (SemEval-2016), pages 1081–1091, San Diego, California. Association for Computational Linguistics.

	Hypernym detection

	Vered Shwartz, Yoav Goldberg, and Ido Dagan. 2016. Improving Hypernymy Detection with an Integrated Path-based and Distributional Method. In Proceedings of the 54th Annual Meeting of the Association for Computational Linguistics (Volume 1: Long Papers), pages 2389–2398, Berlin, Germany. Association for Computational Linguistics.

	Enrico Santus, Frances Yung, Alessandro Lenci, and Chu-Ren Huang. 2015. EVALution 1.0: an Evolving Semantic Dataset for Training and Evaluation of Distributional Semantic Models. In Proceedings of the 4th Workshop on Linked Data in Linguistics: Resources and Applications, pages 64–69, Beijing, China. Association for Computational Linguistics.

	Enrico Santus, Frances Yung, Alessandro Lenci, and Chu-Ren Huang. 2015. EVALution 1.0: an Evolving Semantic Dataset for Training and Evaluation of Distributional Semantic Models. In Proceedings of the 4th Workshop on Linked Data in Linguistics: Resources and Applications, pages 64–69, Beijing, China. Association for Computational Linguistics.

	Hypernym discovery

	Espinosa-Anke, L., Camacho-Collados, J., Delli Bovi, C., & Saggion, H. (2016). Supervised distributional hypernym discovery via domain adaptation. In Conference on Empirical Methods in Natural Language Processing; 2016 Nov 1-5; Austin, TX. Red Hook (NY): ACL; 2016. p. 424-35.. ACL (Association for Computational Linguistics).

	Camacho-Collados, J., Bovi, C. D., Anke, L. E., Oramas, S., Pasini, T., Santus, E., ... & Saggion, H. (2018, June). SemEval-2018 task 9: Hypernym discovery. In Proceedings of the 12th international workshop on semantic evaluation (pp. 712-724).

	Graded hypernym detection

	Vulić, I., Gerz, D., Kiela, D., Hill, F., & Korhonen, A. (2017). Hyperlex: A large-scale evaluation of graded lexical entailment. Computational Linguistics, 43(4), 781-835.

Similar pretraining strategy

Gajbhiye, A., Espinosa-Anke, L., & Schockaert, S. (2022, October). Modelling Commonsense Properties Using Pre-Trained Bi-Encoders. In Proceedings of the 29th International Conference on Computational Linguistics (pp. 3971-3983).

**Paper Topic And Main Contributions:**

This paper presents a novel method of pretraining a BERT-type LM on semantic relations (synonymy, antonymy and hypernymy) so that they perform better in tasks requiring modeling hypernymy. The main idea is to pretrained them so that hyponym-hypernym vector pairs are brought together in the internal representations of the model, whereas noisy samples are pushed far apart. The dataset used as a source for pretraining is WordNet.

The description of this method is very well explained, sound and carefully provides implementation details (even licensing) making this model replicable to the last detail. The paper also comes with a newly created dataset of hypernymy pairs, verbalized and where one of them is masked.

The main limitation I see is that, surprisingly (given the good account of related work), this model is not evaluated on a plethora of works that have proposed datasets and tasks relevant to hypernymy. I find this to be a major limitation of the paper, since instead, it proposes several novel tasks that are conceptually similar or analogous to works that have served as benchmarks for many years.

The second major limitation I can see is that the resulting model still struggles with what intuitively should be an easy task, i.e., not predicting y in the sentence 'y is a type of [mask]'. I find this surprising and, given that the authors hint at applying the proposed pretraining strategy to autoregressive LMs, perhaps these could handle this artifact better.

Finally, in terms of technical contributions, while reporting a sort of negative results in NMoNLI is commendable, it feels more like an interesting piece of analysis instead of a core motivation for the value of this new model.

**Reasons To Accept:**

- New dataset
- Well motivated pretraining objective
- Very well written, and technically sound paper


**Reasons To Reject:**

- Evaluation seems too ad-hoc
- Only using wordnet considering the many taxonomies, lexical databases and thesauri available feels like an important limitation.
- Experimental results are mixed.

**Reproducibility:**

4: Could mostly reproduce the results, but there may be some variation because of sample variance or minor variations in their interpretation of the protocol or method.

**Reviewer Confidence:**

3: Pretty sure, but there's a chance I missed something. Although I have a good feel for this area in general, I did not carefully check the paper's details, e.g., the math, experimental design, or novelty.

---

> ### Author Rebuttal · Authors · 2023-08-29
>
> Thank you for your thoughtful review and suggestions.
>
> **Evaluating our model on the established benchmarks you suggest** was actually something we carefully considered and decided against. Because our central research question was whether or not LM internal representations provide an interface to their observable linguistic behavior, our reasoning was two-fold. First, we wanted our evaluations to reflect extrinsic tasks which had been identified in the literature as points of failure for models like BERT. Second, as our method directly models lexical semantic relations in BERT’s contextual embeddings, we believed such intrinsic evaluations might be positively biased towards our approach without necessarily being indicative of a language model’s ability on extrinsic evaluations. We hope this clarifies our choice of evaluation and makes it seem less ad-hoc. Nevertheless, we agree that these established benchmarks are important to our work, and intend to update our paper with a more comprehensive review of these and our reasoning above for not including these in our evaluation.
>
> **Regarding your point about not predicting y in the sentence 'y is a type of [mask]'.**
> We mention on L466 that our results are consistent with previous findings. We do not believe this to be a major limitation of our approach, but rather of the base language model and pretraining procedure which our work adopts. We agree that autoregressive models can likely better handle these, but believe it is beyond the scope of this work and more appropriate for future work.
>
> **Regarding the negative results on NMoNLI.**
> While the reported average for our method is lower (both methods perform close to chance, 48.9% vs 56.5%), we explain on line 527 and show in Figures 10&11 that these results are likely too high variance to interpret reliably with only 5 samples.

---

### Official Review · Reviewer_WbqA · 2023-08-15

**Soundness:** 2

**Excitement:**

2: Mediocre: This paper makes marginal contributions (vs non-contemporaneous work), so I would rather not see it in the conference.

**Paper Topic And Main Contributions:**

The paper proposes an approach to pre-train large transformer-based language models together with lexical semantic relation (LSR) constraints. Four relations have been considered: i) hypernymy, ii) hyponymy, iii) antonymy, and iv) synonymy. The main idea is to ensure that the related pairs are similar in the four relation-specific vector spaces while pretraining the language models. The paper claims that enforcing these four relation constraints between pairs of words leads to improvements in tasks that require knowledge of hypernymy.

**Questions For The Authors:**

A) If recent large transformer-based LMs still struggle with tasks that require knowledge of hypernymy, could you provide examples instead of just referencing older papers that discuss older LMs?

B) In many places, "BERT (OURS)" is mentioned, but this is confusing. I think BERT+BALAUR is your main proposed approach, not BERT.

C) Is the low MLM loss for BERT+BALAUR in Figure 2 due to BERT+BALAUR having more parameters than BERT(Ours)?

D) Is the improvement achieved by BERT+BALAUR over BERT(Ours) in the prompt completion and NLI tasks because BERT+BALAUR was trained on additional data from WordNET?

E) Why was BERT_large chosen as the base model when RoBERTA_large performs better for hypernymy prediction?


**Reasons To Accept:**

The paper is well-written.

**Reasons To Reject:**

1) I believe the following line in the abstract is not accurate:

"... tasks that require knowledge of hypernymy still pose a challenge for recent pretrained language models (LMs) ..."

Recent LMs like ChatGPT perform well on those tasks. This may be a challenge for older LMs like BERT, but this distinction should have been explicitly mentioned. The way the paper introduces the problem is misleading.

2) It's unclear why pre-training LLMs with LSR constraints is beneficial. One could also fine-tune LLMs with LSR constraints extracted from WordNET. The comparison between the two is missing in the paper.

3) The results do not clearly demonstrate the benefits of the proposed approach. For the prompt completion task considered in the paper, RoBERTA_large outperforms their approach in the Hypernym prediction sub-task.

**Reproducibility:**

3: Could reproduce the results with some difficulty. The settings of parameters are underspecified or subjectively determined; the training/evaluation data are not widely available.

**Reviewer Confidence:**

3: Pretty sure, but there's a chance I missed something. Although I have a good feel for this area in general, I did not carefully check the paper's details, e.g., the math, experimental design, or novelty.

---

> ### Author Rebuttal · Authors · 2023-08-29
>
> Thank you for the detailed review and questions about our work.
>
> **To address your reasons for rejection**, we would first like to clarify that this work was conducted before ChatGPT, and that even if it wasn’t, we strongly believe methods which improve models without relying on ever-increasing amounts of data and compute are important contributions in and of themselves.
>
> It’s no longer surprising that more data and more compute improves performance of LLMs; however there still exist many situations where more data and more compute are not viable solutions (e.g. under-represented languages, or compute-constrained applications).  We believe our work introduces a meaningful potential strategy for such cases.
>
> This also explains why RoBERTa-Large outperforms the models used in our experiments: it was trained on an order of magnitude more data (16GB versus 161GB); which the original authors of RoBERTa also note has a significant impact on downstream performance. We decided to transparently include results for RoBERTa even if they could be interpreted as negative results, because we trust the community to look beyond scale and SOTA; and see our contribution as a method which improves models orthogonally to scaling data or compute.
>
> Rather than demonstrating SOTA, we perform rigorous experimentation that removes potential confounds while demonstrating significant improvements with our method, inline with  [On “Scientific Debt” in NLP: A Case for More Rigour in Language Model Pre-Training Research](https://aclanthology.org/2023.acl-long.477) (Nityasya et al., ACL 2023).
>
> **However, we agree that the claim in the abstract is no longer accurate, or at least too ambiguous**, and that we should be more specific than “recent pretrained language models”, in addition to clarifying considerations of scale, as above. This edit and discussion is straightforward to incorporate and will be done for the camera-ready version if the paper is accepted.
>
> **Regarding the issue of finetuning which you raise**, we explored the option but were not able to resolve issues of catastrophic forgetting; even when interleaving the original pretraining objective. As all the related work we found on incorporating lexical semantics into LMs involved pretraining rather than finetuning, we adopted a similar approach. We believe this is a potential direction for future work rather than a fundamental limitation of this work, especially given the similar approach of previous published works on this topic. However, we agree this should be explicitly discussed and justified rather than glossed over, and can do so.
>
> **To answer your questions**:
>
> **A.** Please see the discussion above.
>
> **B.** This is explained on line 378. To remove any confounds such as differences in training data, model implementation, and random seed induced differences, we pretrain two versions of the exact same 24h-BERT model: one with and without Balaur, keeping everything else the same. We use BERT(OURS) to distinguish from published checkpoints of BERT-Large.
>
> **C.** For all our evaluations, BERT+BALAUR and BERT(OURS) have the exact same architecture and number of parameters.  Balaur parameters are only used to compute a complementary loss function during pretraining, but are not part of the computation graph of MLM or any of our evaluations.
>
> **D.** Exactly, that was the hypothesis we were validating: that we could successfully incorporate knowledge (data from WordNet) into a model by modulating its latent representations. Importantly, there are no additional training examples; both BERT+BALAUR and BERT(OURS) see the exact same texts throughout pretraining; we use the data from WordNet to simultaneously compute a complementary loss without requiring additional forward or backward passes.
>
> **E.** See A and B.

---

### Official Review · Reviewer_AUBR · 2023-08-18

**Soundness:** 4

**Excitement:**

3: Ambivalent: It has merits (e.g., it reports state-of-the-art results, the idea is nice), but there are key weaknesses (e.g., it describes incremental work), and it can significantly benefit from another round of revision. However, I won't object to accepting it if my co-reviewers champion it.

**Paper Topic And Main Contributions:**

The paper introduces a novel neural architecture that learns to transform output hidden states so that knowledge about hypernymy, hyponymy, and antonymy can be injected. They train a BERT model with this, and evaluate against a self-trained BERT, as well as BERTlarge, for language modelling as well as a variety of tasks specific to lexical semantic relations. They evaluate very thoroughly and find that their method generally improves performance on these specific tasks,

**Questions For The Authors:**

Why were the prompts chosen in this way? Why would the prefix "in the context of" be necessary, and did you test without it? How did this affect the results?

**Reasons To Accept:**

This paper is well written and a lot of work has clearly been completed. The analysis is thorough and done in a variety of settings and tasks. All necessary details are there, and particularly the discussion of results is clear, fair, and refers to previous work.

The paper addresses a weakness of language models in a targeted way, and I think that the architecture may be applicable to the injection of other kinds of knowledge as well.

**Reasons To Reject:**

My main reason to reject this work would be that I'm not confident it has broad applicability. I have no doubts about the positive results for prompt completion and monotonicity, I'm not sure that the improved result on language modelling warrants such a complex architecture. I'm sympathetic with the author's constraints in terms of computational power and it makes sense that they therefore tested this on BERT, but it doesn't follow that larger models would still benefit from supervision like this, which makes my doubts about the broader impact of this work even larger. I however admit that it is possible that I'm underestimating this, as I'm not too familiar with the area.

**Reproducibility:**

4: Could mostly reproduce the results, but there may be some variation because of sample variance or minor variations in their interpretation of the protocol or method.

**Reviewer Confidence:**

2: Willing to defend my evaluation, but it is fairly likely that I missed some details, didn't understand some central points, or can't be sure about the novelty of the work.

---

> ### Author Rebuttal · Authors · 2023-08-29
>
> Thank you for your careful and considerate review of our work.
>
> **To address your main reason for rejection**, we would like to clarify the main intended contribution of our work. The improvement in language modeling or obtaining SOTA was not our goal; rather we wanted to verify our hypothesis that knowledge of lexical semantics (or more precisely, *improvements on tasks requiring knowledge of hypernymy*) could be instilled in a language model by controlling its latent representations (rather than by e.g. providing it additional text to pretrain on or increasing its number of parameters). The observed improvement in language modeling performance was an interesting and unanticipated positive result, but not directly related to our original research question. We see the positive results for prompt completion and monotonicity as the results which most directly support our original hypothesis.
>
> We agree it is unclear whether our approach generalizes to larger models, however we believe that models at the scale of BERT-Large are still useful and used in a variety of contexts; and that the generalizability of our method to even larger models makes for an interesting future research direction rather than a fundamental limitation of our work.
>
> **To answer your questions**, we added “in the context of hypernymy” as a prefix based on the observation that standard prompts like “An [X] is a [MASK]” or “An [X] is a type of [MASK]” could have valid completions unrelated to hypernymy, e.g. synonyms, which we wanted to disambiguate. Without this prefix, improvement from our approach persisted, although performance fell across all models. As we believe our use of a disambiguating prefix is better suited for our evaluation, and found no additional insights from these prefix-less results, they were not included.

---

### Meta-Review · Area_Chair_7WWE · 2023-09-24

**Recommendation:** 5

**Metareview:**

The paper introduces a novel BERT-type pre-training model that learns to transform output hidden states with knowledge about hypernymy, hyponymy, and antonymy (from WordNet) injected. They trained the model from scratch on English Wikipedia and BookCorpusOpen, comparing to BERT. The proposed method is interesting, evaluated on prompt completion and monotonicity NLI (MoNLI). The amount of improvements while did not impress some of the reviewers, have be in reasonable or expected range of improvement as similar research that also attempts modifications for BERT-type of encoder models. At the same time, this work has demonstrated novelty and technical contributions that *ACL/EMNLP conferences are looking for, offers a good reference point for future work on pre-training models with external semantic knowledge, and should be encouraged for publication.

---

### Decision · Program_Chairs · 2023-10-07

**Decision:**

Accept-Findings

**Comment:**

The paper introduces a novel BERT-type pre-training model that learns to transform output hidden states with knowledge about hypernymy, hyponymy, and antonymy (from WordNet) injected. They trained the model from scratch on English Wikipedia and BookCorpusOpen, comparing to BERT. The proposed method is interesting, evaluated on prompt completion and monotonicity NLI (MoNLI). The amount of improvements while did not impress some of the reviewers, have be in reasonable or expected range of improvement as similar research that also attempts modifications for BERT-type of encoder models. At the same time, this work has demonstrated novelty and technical contributions that *ACL/EMNLP conferences are looking for, offers a good reference point for future work on pre-training models with external semantic knowledge, and should be encouraged for publication.